# Extraction Methods of Microplastics in Environmental Matrices: A Comparative Review

**DOI:** 10.3390/molecules30153178

**Published:** 2025-07-29

**Authors:** Garbiñe Larrea, David Elustondo, Adrián Durán

**Affiliations:** 1BIOMA Institute for Biodiversity and the Environment, University of Navarra, Irunlarrea 1, 31008 Pamplona, Spain; delusto@unav.es; 2School of Sciences, Department of Chemistry, University of Navarra, Irunlarrea 1, 31008 Pamplona, Spain

**Keywords:** environmental matrices, microplastics, extraction, organic matter, density separation

## Abstract

Due to the growing issue of plastic pollution over recent decades, it is essential to establish well-defined and appropriate methodologies for their extraction from diverse environmental samples. These particles can be found in complex agricultural matrices such as compost, sediments, agricultural soils, sludge, and wastewater, as well as in less complex samples like tap and bottled water. The general steps of MPs extraction typically include drying the sample, sieving to remove larger particles, removal of organic matter, density separation to isolate polymers, filtration using meshes of various sizes, oven drying of the filters, and polymer identification. Complex matrices with high organic matter content require specific removal steps. Most studies employ an initial drying process with temperature control to prevent polymer damage. For removal of organic matter, 30% H_2_O_2_ is the most commonly used reagent, and for density separation, saturated NaCl and ZnCl_2_ solutions are typically applied for low- and high-density polymers, respectively. Finally, filtration is carried out using meshes selected according to the identification technique. This review analyzes the advantages and limitations of the different methodologies to extract microplastics from different sources, aiming to provide in-depth insight for researchers dedicated to the study of environmental samples.

## 1. Introduction

Plastic pollution has become one of the most pressing environmental issues affecting global health [1,2,3]. Plastics are synthetic, ubiquitous, and non-biodegradable compounds that tend to accumulate in the environment [4,5]. Although it is difficult to provide precise estimates of annual plastic pollution due to variability among sources, an exponential increase in production has been observed over recent decades from 2 million metric tons (MT) in 1950 to nearly 450 million MT currently [6,7,8,9]. If this trend continues, global production is projected to reach approximately 13 billion MT by 2050 [2,7]. Of all the plastic produced, 9% is recycled, 12% is incinerated, and the remaining 79% accumulates in landfills and aquatic and terrestrial ecosystems [5]. These values are particularly concerning: an estimated 4.8 to 12.7 million MT of plastic end up in marine ecosystems, while around 1.5 million MT accumulate in terrestrial environments [5,6]. Moreover, it is projected that, by 2050, nearly 70 million MT of plastic will be released into the environment [3].

Plastic materials, especially those exposed to environmental conditions, are subject to various physical, chemical, mechanical, and biological degradation processes, breaking down into microscopic particles smaller than 5 mm in their longest dimension, known as microplastics (MPs) [1,9]. While their size classification is not yet standardized, a general proposal distinguishes: nanoplastics (<1 mm), MPs (1–5 mm), mesoplastics (5–20 mm), macroplastics (>20 mm), and megaplastics (>100 mm) [4,8]. Other authors define MPs as ranging from 1 µm to 5 mm [9,10], some differentiate between large MPs (1–5 mm) and small MPs (0.5–1 mm) [11], while most agree that MPs are particles smaller than 5 mm [1,2,3,5,7]. According to their origin, MPs are classified as primary or secondary. Primary MPs, also known as microbeads, are intentionally manufactured for use in cosmetic, detergent, and textile products, among others [12,13,14,15]. Secondary MPs result from the degradation of macroplastics due to environmental weathering processes [2,3,5,8,9]. These are the most abundant type, representing over 90–95% of MPs present in the environment [2,5]. Morphologically, MPs can appear as fibers, fragments, granules, pellets, spheres, films, flakes, or foam, fragments being the most prevalent [2,4,7]. They also exhibit a wide range of colors and chemical compositions. Although approximately 45 different plastic types have been identified in the environment, the most commonly detected include polypropylene (PP), polyethylene (PE), polystyrene (PS), and polyethylene terephthalate (PET) [7,8,16].

The agricultural sector is the largest contributor to MPs generation. MPs are produced during agricultural activities (plastic films for mulching or greenhouse coverings) and also during food processing through the release of microparticles from packaging materials during handling and consumption. In this context, the primary concern stems from single use or disposable packaging, which accounts for approximately 40.5% of annual plastic demand in Europe [17].

Regarding the accumulation of MPs, it is difficult to provide precise data, since they originate from the degradation of macroplastics, and the parameters vary considerably. However, a study conducted by Zhang et al. [18] analyzed the concentration of MPs in Chinese soils using a machine learning model, which allows the spatial distribution of MPs in soil to be studied from 1980 to 2018. The results showed an increase from 45 MPs/kg of soil to 1156 MPs/kg of soil, respectively. According to the study, 39% of this accumulation comes from industry, 30% from agricultural films, 17% from tire waste, and 14% from domestic waste. The authors also point out that, by the year 2050, this pattern is expected to stabilize due to the implementation of regulatory measures.

The exponential growth in the number of publications related to this research topic in recent years (Figure 1) highlights the need for this review, which provides an updated overview of the most recent methodologies for MPs extraction, classifying them according to the type of environmental matrices and, in particular, based on the percentage of organic matter (OM) present, a determining factor in the effectiveness of the process (see Section 3). To generate Figure 1, a literature search was conducted using the Web of Science database. The search combined the terms “microplastics” and “extraction” with keywords corresponding to each sample type: “sludge,” “compost,” “agricultural soils,” “sediments,” “wastewater,” “atmospheric samples,” “beach sand,” “seawater,” “freshwater,” and “drinking water,” using the AND operator (e.g., “microplastics AND extraction AND sludge”). These queries were performed individually for each sample type. The retrieved publications were then grouped according to the percentage of OM typically associated with each matrix. The search was restricted to research articles published between 2016 and 2024, as very few relevant publications were available prior to 2016. To the best of our knowledge, there is no existing review that comprehensively covers all types of environmental matrices, as this work does. Most studies focus on the extraction process from a single sample type, particularly from agricultural soils [1,19,20,21,22,23]. Others concentrate on specific matrices such as sediments [24,25,26] or fresh and seawater [27,28], among others. The accurate extraction of polymers from environmental matrices is a critical step, as, if not performed correctly, it can lead to cross contamination, detection errors, or inadequate identification of MPs. The general extraction steps are described, and the advantages and limitations of using different reagents in both removal of organic matter (ROM) and density separation (DS) processes are analyzed, emphasizing the need to achieve a balance between cost, effectiveness, time, and toxicity. This review aims to serve as a key for researchers beginning to study the extraction of MPs from environmental matrices, however, it remains necessary to progress toward a standardized methodology that can be applied to different types of samples without compromising key parameters such as temperature, time, cost, or chemical safety.

## 2. Steps of Extraction Process

Samples from the agricultural industry include sands, sludge, compost, sediments, wastewater, soils, etc. These samples generally require pretreatment to extract the MPs present in them. The extraction of MPs depends both on the type of sample and the technique used for their subsequent identification. The more complex the sample, the more preparation is needed [3,29,30,31,32,33,34]. The extraction process includes several steps common to any type of sample (with some variations): initial drying of the sample, sieving using different mesh sizes, ROM, isolation of polymeric particles by DS, filtration, final drying of the filters, and identification [3,35,36] (Figure 2).

If the samples contain large particles, such as leaves, branches, or sticks, it is recommended to remove them before analysis. They can be manually removed using stainless-steel tweezers [11,37,38,39,40], or, alternatively, some authors choose to filter the sample using a column of sieves with different sieve sizes. The retained particles (>5 mm) are discarded since they are not considered MPs [37,41,42,43].

### 2.1. Removal of Organic Matter (ROM)

The purpose of the ROM is to eliminate organic matter from samples, particularly in those with high OM content. This is typically achieved using one or more chemical agents—oxidizing, alkaline, acidic, or enzymatic. Among these, hydrogen peroxide (H_2_O_2_) is the most commonly used reagent, as shown in Table 1. According to Rede et al. [36], 76% of the reviewed studies on environmental samples employed H_2_O_2_ for OM removal. Similarly, a recent review by Sahai et al. [44] reported that, in studies focused on agricultural soils, H_2_O_2_ was used in 53.1% of the cases. Fenton’s reagent (a mixture of H_2_O_2_ and a ferric ion catalyst, such as Fe(II)SO_4_) was applied in 20.4% of the studies. Another 20.4% did not apply any ROM treatment, while 6.1% used alternative reagents not specified in the main categories. The typical procedure involves adding the reagent to the sample, stirring it under controlled temperature to facilitate the reaction, and then allowing the mixture to settle.

In some types of samples, such as sediments, the addition of H_2_O_2_ generates bubbles and foam due to the release of CO_2_, so the reagent must be added gradually. In other cases, the process must be repeated several times to completely remove the present OM [34,43,45,46]. Schrank et al. [47] evaluated the effect of various reagents commonly used in ROM on eight common polymers, with the aim of deciding whether these chemicals altered the morphology of the plastics, degraded them, or dissolved completely. In their study, they tested NaOH, HNO_3_, H_2_O_2_, and Fenton’s reagent. The results showed that strong acids and bases are highly destructive to most polymers, and therefore their use is not recommended. Although H_2_O_2_ caused some degradation in polyamide, no adverse effects were observed on any of the analyzed polymers when treated with Fenton’s reagent. However, for more effective OM removal, it is recommended to combine Fenton’s reagent with enzymatic digestion. Hurley et al. [48] mention that, while the use of H_2_O_2_, NaOH, and KOH showed signs of degradation in the polymers, this did not occur with Fenton’s reagent. Nevertheless, it is important to emphasize the need to control the temperature below 40 °C and maintain a pH value of 3 [48,49]. Meanwhile, Munno et al. [50] consider oxidation with H_2_O_2_ as an effective method for the digestion of marine samples, water, and sediments, providing that the temperature does not exceed 60 °C to avoid MPs degradation. A recent article by Peneva et al. [38] notes that methods such as Fenton’s reagent or strong acidic and alkaline digestions can degrade certain polymers. In contrast, enzymatic digestion, although less aggressive and less likely to damage MPs, does not efficiently remove all the OM present in the sample. Another study by Gulizia et al. [51] indicates that, when analyzing PS, both NaOH (10 M) and KOH (1.8 M) at <90 °C cause minimal degradation of the physical and chemical properties of the polymers. In contrast, HNO_3_ (15.8 M) causes degradation, swelling, and nitration of the polymer, while H_2_O_2_ generates mild effects. Regarding NaClO, Hagelskjær et al. [52] state that it is more efficient in ROM than H_2_O_2_: with NaClO at 50%, a 99% elimination of OM is achieved, compared to 28% with H_2_O_2_ at 30% and 75% with the Fenton reagent. However, NaClO concentrations higher than 12.5% by volume can degrade and deform polymers such as PA and PET (Table 1).

In summary, there is great variability in the reagents used for the removal of OM, and further research is needed. Additionally, it is crucial to consider the nature of each sample, as factors such as origin or OM content influence the choice of appropriate treatment. If the sole objective is to detect the presence of MPs, slight degradation would not be an issue; however, if the goal is to analyze individual particles, it would be recommended to avoid it.

The most used reagents in the ROM process and their respective characteristics are provided in Table 1. The differences between them include matrix digestion efficiency, level of aggressiveness and damage to the polymer, method complexity (number of steps, complicated and expensive reagents, reproducibility), and time required [47].

**Table 1 molecules-30-03178-t001:** Most commonly used reagents in the ROM process and some of their characteristics.

Process	Reagent	Characteristics	References
Oxidative	H_2_O_2_	Often used at 30–35% by volume. Removes OM with high efficiency. Slight effect on some polymers. Long and temperature-dependent analyses.	[34,36,43,45,46,47,50,52,53,54,55]
	Fenton’s reagent (H_2_O_2_+Fe(II)SO_4_)	Effective for samples with high OM content. Temperature-dependent. pH adjustment to 3–5 is necessary. No effect on polymers. Shorter analyses than with H_2_O_2_.	[38,47,49,52,53,54,56]
Alkaline	NaOH KOH NaClO	Useful for biological samples. Less effective in removing OM from environmental samples. Affects polymers.	[38,47,51,52,53,54]
Acid	HNO_3_ HCl HClO_4_ H_2_SO_4_	Effective in removing OM but digests some MPs. Often combined with alkaline digestion.	[38,47,51,53]
Enzymatic	Cellulase Lipase Protease Proteinase K	Mild method. Does not affect MPs. Often complemented with a subsequent oxidation. Expensive and costly (several enzymes are used). Not recommended for environmental samples.	[38,53,54,57]

The isolation of MPs consists of separating them from the sample. In most cases, it is carried out through DS. This method is based on the flotation of particles in a saturated saline solution with a density higher than that of the MPs present in the sample. In this way, MPs rise to the surface of the mixture and are filtered and collected, while the denser fraction settles at the bottom and is discarded [36].

Salts such as NaCl, ZnCl2, CaCl2, and NaI are commonly used, NaCl being the most widely used reagent in most cases (Table 2). Dorau et al. [45] and Rede et al. [36] mention that 63% of the reviewed studies use a single saline solution, 23% combine two consecutive saline solutions, and, in only one article, three saline solutions are used. On the other hand, the review by Sahai et al. [44] points out that 32% of the reviewed studies use NaCl, followed by ZnCl_2_ in 15%, 23.4% use several successive saline solutions, 10.6% use H_2_O, 8.5% use NaI, and 4.3% use NaBr, while CaCl_2_ is the least used reagent.

The choice of reagent highly depends on the polymers present in the sample. NaCl is an easy to obtain, inexpensive, non-toxic reagent and is effective in separating most common plastics; however, its maximum density is 1.2 g/cm^3^, which is insufficient to separate polymers such as PET or PVC that have higher densities (~1.4 g/cm^3^). In these cases, other salts such as ZnCl2 or NaI are required, because they can generate solutions with densities of 1.50–1.80 g/cm^3^ and 1.55–1.80 g/cm^3^, respectively, thus covering a broader range and ensuring the separation of all polymers present in the sample [58,59]. However, ZnCl2 and NaI are very expensive and toxic salts. Nevertheless, in the case of ZnCl_2_, it has been shown that costs can be reduced by reusing the solution [60,61]. Rodrigues et al. [60] develop a method that allows the salt to be reused up to six times while maintaining recovery rates above 95% in all cases. CaCl2 is a more economical alternative with a density slightly higher than that of NaCl (1.30–1.35 g/cm^3^). However, it leaves many residues in the samples, requiring thorough washing to avoid interferences, thus increasing the analysis time. The density difference between both reagents is not significant enough to be a deciding factor; however, the additional cleaning required when using CaCl_2_ may be a key consideration [38,45,62,63,64].

In addition, CaCl_2_, due to its high viscosity, tends to generate flocculation, which limits its use [65]. Other reagents such as NaBr, ZnBr_2_, or sodium polytungstate (SPT) have been used in some studies, although less frequently. SPT is a more expensive reagent than NaCl, but it allows the preparation of solutions with very high densities (up to 3.1 g/cm^3^) without safety issues [58]. NaBr presents some toxicity [45]. Jing et al. [66] used a NaCl and NaI mixture (1:1) for MPs extraction while Liu et al. [67] combined NaBr and ZnCl_2_ and Kim et al. [68] used a ZnCl2:CaCl2 mixture (2:1.4), achieving recovery efficiencies close to 90% in all three cases. The choice of the most suitable reagent involves finding a balance between cost, toxicity, and effectiveness in separating the polymers present in the sample, considering that the price of salts tends to increase as the achievable density increases [45].

A summary of the density ranges of the most common polymers, along with the densities of the most commonly used reagents in the DS process, is provided in Table 2.

Alternatively, oil extraction methods have been developed, which combine a low-density oil with the oleophilic properties of plastics, allowing MPs to accumulate in an oil layer above the aqueous solution. Several studies have tested this method using different types of oil: canola oil, achieving a mean recovery efficiency of 84% [69], olive oil, reporting recoveries of 80% [70], and sunflower oil, obtaining recovery rates between 82% and 98% [71].

**Table 2 molecules-30-03178-t002:** Most common polymers, reagents, MPs extracted in the DS process and their references, where ***** refers to spiked MPs and ***** refers to MPs from environmental matrices.

Polymers and Their Density (g/cm^3^) [62,72]	Salts and Their Density (g/cm^3^) [54,59,63]	MPs Extracted	References
PP (0.85–0.92) HDPE (0.94–0.98) LDPE (0.89–0.93) PS (1.04–1.06) PMMA (1.14–1.20) PA, Nylon (1.12–1.24) PC (1.20–1.22) PU (1.20–1.26) PES (PETG) (1.27) PET (1.37–1.41) PVC (1.38–1.45)	NaCl (1.15–1.30)	PP	[39] ******, [40] *****, [69] *****, [73] ******, [74] *****
		HDPE	[23] ******, [31] *****, [39] ******, [40] *****, [61] *****, [69] *****, [73] ******, [74] *****
		LDPE	[23] ******, [31 *****], [39] ******, [40] *****, [61] *****, [69] *****, [74] *****
		PS	[39] ******, [69] *****, [73] *****, [74] *****
		PA	[39] ******
	CaCl_2_ (1.30–1.35)	PP	[42] *****
		HDPE	[42] ******
		LDPE	[32] *****, [42] ******
		PA	[32] *****, [42] *****
		PC	[42] *****
		PU	[42] *****
	SPT (1.4–1.70)	PP	[75] *****
		HDPE	[75] *****
		LDPE	[75] *****
		PS	[75] *****
		PES	[76] *
	NaI (1.55–1.80)	PP	[46] *****, [77] *****
		HDPE	[46] *****, [77] ******
		LDPE	[77] ******
		PS	[46] *****, [77] ******
		PMMA	[46] *****
		PA	[46] *****, [77] *****
		PET	[46] *****
		PVC	[46] *****, [77] *****
	ZnCl_2_ (1.50–1.80)	PP	[38] *****, [69] *, [78] *****, [79] *****
		HDPE	[38] *****, [69] *, [70] ******, [78] *****, [80] *****, [81] *****
		LDPE	[38] *****, [69] *, [70] ******, [78] *****, [80] *****, [81] *****
		PS	[38] *****, [69] *, [70] ******, [78] *****
		PMMA	[78] *****
		PA	[29] *****, [38] *****, [78] *****
		PU	[78] *****
		PES	[29] *****
		PET	[29] *****, [38] *****, [69] *, [70] ******, [78] *****, [79] *****, [80] *****, [81] *****
		PVC	[29] *****, [38] *****, [69] *, [70] ******

It is also important to note that the extraction processes of the sample, especially in the final filtration steps, largely depend on the following characterization and identification technique to be used.

For instance, when using laser direct infrared imaging (LDIR), it is advisable to filter the sample directly onto a gold filter (or aluminum as an alternative) of appropriate diameter. This allows the filter with the retained MPs to be directly inserted into the device for analysis, as demonstrated by Ghanadi et al. [42] and Cheng et al. [82]. This type of filter is required because LDIR relies on a reflective surface. In contrast, filters made of cellulose or other polymers can interfere with the analysis and introduce cross-contamination, compromising the reliability of the results. Since this technique is particularly sensitive to residual organic and inorganic matter, the extraction protocol must be thoroughly optimized. This includes repeating the ROM process until the filters are completely clean and performing an appropriate DS to maximize the microplastic recovery rate. Additionally, it is recommended to conduct experimental blanks to assess potential contamination from air, laboratory materials, or sample handling [42,82].

Raman spectroscopy is another widely used technique for the analysis of microplastics in environmental samples [1,29,38,64,71]. This non-destructive technique allows the identification of polymers based on their chemical structure and composition, offering high sensitivity even with minimal sample amounts. One of the main challenges of Raman spectroscopy is unwanted fluorescence that may occur in the presence of organic and inorganic contaminants. This fluorescence interferes with the Raman signal, making polymer identification difficult. Therefore, it is essential to apply an effective pretreatment that removes as many impurities as possible before analysis [1].

Garzón-Vidueira et al. [71] emphasized that, when oils are used during DS, selecting an appropriate washing solvent is critical for reducing background noise. The best results were obtained using ethanol in combination with sunflower oil, which significantly reduced background noise. Likewise, Crutchett et al. [29], working with sediment samples, highlighted the importance of using stainless-steel small-diameter filters, which help concentrate MPs into a smaller area, thereby facilitating detection and analysis.

If the characterization technique involves morphological analysis—such as optical microscopy—it is important to consider that aggressive reagents may degrade the polymers or break them into particles too small to detect. Temperatures above 60–70 °C should also be avoided to prevent thermal degradation. Moreover, excessive agitation during treatment should also be minimized, as it may contribute to particle fragmentation [1].

### 2.2. The Correlation Between OM Amount and Abundance of MPs

Extraction methods could be classified according to the type of sample to be analyzed, as its composition influences the choice of procedure. The percentage of OM in a sample represents the amount of organic carbon present that can be of natural or anthropogenic origin. That of natural origin comes from the decomposition of plants and animals, such as leaves, branches, and twigs, and highly decomposed forms such as humus. The greater the level of decomposition of the OM in the sample, the higher the amount of carbon and, consequently, the higher the % of OM. Among the anthropogenic sources, accidental spills or releases that may occur stand out; however, they represent a lower percentage compared to naturally derived OM.

To carry out analysis of the OM of samples, these are destroyed by chemical or thermal methods, releasing CO2, which will subsequently be measured and transformed into the total organic carbon percentage (%TOC) [83]. Various authors have reported a strong correlation between %TOC and the abundance of MPs [67,84]. However, few studies on MPs analyze this data before starting their analyses. Some samples, such as sludge and compost, have a high OM content, others such as sediments and soils have a moderate OM content, while drinking water presents a minimal amount. Regarding the OM content present in sludge samples, this depends, among other factors, on the procedure employed in wastewater treatment plants (WWTPs). The %OM in dry weight sludge samples usually ranges between 50 and 70% [85]. Some authors provide specific values in their research, such as Hurley et al. [48] and Wang et al. [86], who reported 51.5% OM, while Al-Azzawi et al. [87] reported a value of 70.5%. Ragoobur et al. [39] studied the %TOC in three soils of different composition used for vegetable cultivation, obtaining values between 0.32 and 0.66%. Another recent study conducted by Seo et al. [23] analyzed ten samples from different agricultural soils, with TOC results ranging from 0.76 to 2.87%. Simon-Sánchez et al. [34] analyzed sediment samples from deltas, obtaining TOC values between 3.6 and 4.9%. Bailey et al. [88] also analyzed sediment samples from bays and obtained percentages between 0.8 and 2.2%, while Liu et al. [67] analyzed marine sediments and reported values ranging from 0.19 to 1.6%. These differences in the percentage ranges are due to factors such as the depth at which the sample was taken, salinity, sample location, and distance to rivers (among others), as, generally, %TOC values are higher the closer the sample is to a river mouth. With regard to drinking water samples, the ideal TOC value is below 5 mg/L. If it exceeds 6 mg/L, corrective measures must be applied, and if it reaches or exceeds 7 mg/L, the water is considered unsafe for human consumption [89]. In their study, Zeeshan et al. [90] analyzed the content of dissolved OM in drinking water samples, obtaining ranges between 0.3 and 5.7 mg/L, which corresponds approximately to TOC values between 0.00003 and 0.00057%. The average concentration obtained was 1.6 mg/L.

The most common types of environmental samples, along with their classification based on OM content, are shown in Figure 3. This classification is introduced in this review and is considered a useful tool for distinguishing between different extraction methodologies.

## 3. Extraction of MPs from Environmental Samples

### 3.1. Samples with High OM Content (Sludge and Compost)

#### 3.1.1. Sludge

Ragoobur et al. [39] carried out the extraction of MPs from sludge samples. First, the samples were mixed with distilled water and stirred for 30 min, then left to settle for 24 h. Next, a NaCl saline solution was added and mixed with the sample for 15 min, followed by a 2 h settling period. The resulting suspension was centrifuged for 20 min at 1500 rpm. Subsequently, the OM was removed by adding 30% H_2_O_2_ for 7 days at room temperature. Finally, the sample was vacuum filtered using a 0.45 µm nitrocellulose filter. A recent study by Yli-Rantala et al. [75] addressed the extraction of MPs from primary and biological sludge from the paper industry. To remove OM, a chemical–enzymatic digestion was carried out to degrade the cellulose fraction of the sample. The sludge samples were diluted with ultrapure water and filtered using a stainless-steel filter of 26 µm (Ø 90 mm). The retained solids were immersed in a 1% sodium dodecyl sulfate (SDS) solution for 24 h. Subsequently, filtration was repeated with a Ø 45 mm filter, and the filter with the retained solids was treated with 30% H_2_O_2_ for 72 h. Afterward, an enzymatic digestion with cellulase in acetate buffer was carried out for 96 h, followed by a second oxidation with 30% H_2_O_2_ for 72 h. In the case of primary sludge, a DS was performed using an SPT solution, with a settling time of 48–72 h at room temperature. Finally, the supernatant was filtered using a 26 µm filter (Ø 90 mm). In this type of sample, saturated solutions of ZnCl_2_ are also commonly used, following a procedure very similar to those previously mentioned: an initial filtration, digestion with 30% H_2_O_2_, DS, a second filtration, and drying of the filters at room temperature for 24 h [78].

#### 3.1.2. Compost

For the extraction of MPs from compost samples, an innovative method was employed, consisting of a switchable CaCl_2_ density separation column that avoids the use of aggressive chemical treatments. In the process, the sample was mixed with ultrapure water by sonication, using an ice bath to prevent overheating. Next, the CaCl_2_ solution was added along with ten drops of Lutensol TO7 as a surfactant, and the mixture was left to settle for 2 h at 60 °C. After this process, progressive cooling was carried out: first to 14 °C for 1 h and 45 min, then to 11 °C for 15 min (with 30 s of stirring), and finally, it was left to rest at 11 °C for 1 h without stirring, allowing the column to solidify. Once solidified, the upper 5 cm, where the MPs are located, was cut off, and the procedure was repeated three times. Finally, the cut fragment was melted by adding CaCl_2_ at 60 °C [32]. Fosu-Mensah et al. [35] analyzed organic soil amendments using compost samples from municipal solid waste. To remove the OM, the sample was treated with 30% H_2_O_2_ and distilled water, heating it to 60 °C for 2 h. Subsequently, it was left in an oven at 60 °C for 24 h. For DS, a ZnCl_2_ solution was used, which was added to the samples and vigorously stirred for 5 min, then left to settle overnight. Finally, the supernatant was filtered, and the filters were dried in an oven at 50 °C for 15–20 min. The last of the reviewed studies used an extraction method on solid samples. First, a wet peroxide oxidation (WPO) was performed by mixing 30% H_2_O_2_ with Fenton’s reagent. This mixture was added to the samples along with ultrapure water, which allowed temperature control without the need for ice baths. The mixture was left to rest at room temperature for 6 h and then at 35 °C overnight. Subsequently, daily additions of H_2_O_2_ were made so that the digestion process would progress gradually, ending when all the organic material had settled and the solution was no longer turbid. This process can last between 2 and 10 days. For DS, two solutions were used: first ultrapure water and then NaI. In addition, a sediment–microplastic isolation (SMI) unit was used, a device designed to isolate MPs and widely recommended in all MPs extraction methods that include DS [46].

As a summary of Section 3.1, in the treatment of sludge for MPs extraction, it is common to begin with a DS phase, typically using saline solutions such as NaCl, ZnCl_2_, or SPT. This stage is usually accompanied by prolonged settling processes, which can extend up to 72 h, and complemented by centrifugation. For ROM, the most commonly used reagent is 30% H_2_O_2_, applied in multiple cycles lasting from 12 h to several days, depending on the concentration of OM present in the sample. In certain cases, this treatment is complemented with enzymatic digestions, such as cellulase, or with SDS to dissolve proteins and lipids. The final stages often include successive filtrations using metal or nitrocellulose filters, with pore sizes ranging from 0.45 µm to 26 µm. Some protocols also include air drying of the filters prior to final analysis. In compost samples, combinations of chemical digestion and DS are employed. ROM is typically carried out with 30% H_2_O_2_, sometimes combined with Fenton’s reagent, in treatments that may extend over several days. For DS, saturated solutions of ZnCl_2_, NaI, or CaCl_2_ are mainly used. The latter has been applied in innovative methods that avoid the use of hazardous reagents by using thermally controlled switchable columns. Some protocols also integrate specialized methods such as SMI, designed to optimize microplastic recovery.

### 3.2. Samples with Moderate OM Content (Agricultural Soils, Sediments, and Wastewater)

#### 3.2.1. Agricultural Soils

In most studies of agricultural soils, two general steps are used: ROM and DS. Isari et al. [31] used a NaCl saline solution for 5 min with agitation at 200 rpm and a settling time of 2 h. The supernatant was then passed through a 500 µm filter and subsequently through a 20 µm filter. Finally, the OM was removed with 30% H_2_O_2_ at 70 °C for 2 days. In some cases, a very similar method is used but with the addition of an initial centrifugation to facilitate the DS of the polymers and a final centrifugation to remove residual NaCl [39]. Other studies have also emphasized the importance of centrifugation in the extraction of MPs from soil samples. Pfohl et al. [76] used an SPT solution subjected to centrifugation at 30,000 rpm for 1 h at 25 °C for the DS step, then, the vials were frozen at −20 °C overnight, and the top 2 cm was cut off, where MPs concentrate. The bottom part of the vial, where sediments are found, was also cut and subjected to a second centrifugation to recover any trapped MPs. The density was then adjusted to 1.6 g/cm^3^ with distilled water, followed by a final centrifugation, and, finally, the samples were frozen again overnight, cut again, and prepared for final analysis.

Radford et al. [69] compared DS using different reagents such as NaCl, ZnCl_2_ solutions, and canola oil, the latter taking advantage of the oleophilic properties of some polymers that migrate to the oil layer during separation. They are not the only ones to use oil-based separation; Prosenc et al. [70] used olive oil for polymer separation. In their procedure, after separation was achieved, the sample was frozen at −18 °C and then ejected using a piston, retaining the oily layer, which was later melted.

A recent study by Rede et al. [36] applied green analytical chemistry (GAC) for MPs extraction. First, the samples were dried for 48 h at 40 °C. Then, a pretreatment with four sieves of different sieve sizes (5 mm, 1 mm, 100 µm, and 40 µm) was performed. Subsequently, 15% H_2_O_2_ was added for 15 min, and the sample was placed in a water bath at 40 °C for 3 h. A NaCl solution was then used, followed by centrifugation for 30 min at 4500 rpm. The supernatant was filtered with a 1.2 µm glass fiber filter, and this DS process was repeated three times. Finally, the three filters obtained were dried at 40 °C overnight. Some authors mention in their studies that, for soil samples, OM should be removed with 30% H_2_O_2_, Fenton’s reagent, or enzymatic–oxidative digestion, depending on the OM content. When using H_2_O_2_, it should be applied at 30%, 60 °C, and for 44 h until bubbling ceases. For DS, they tested NaCl, SPT, NaBr, and ZnCl_2_ solutions, concluding that it is necessary to balance cost and effectiveness. They highlighted that, in general, higher-density solutions are more suitable as they allow very high recovery rates, especially when using ZnCl_2_ [45].

Forsythe et al. [41] evaluated elutriation as a pretreatment before extraction. First, the samples were dried at 50 °C, sieved through a Ø 4.75 mm sieve to remove large debris, and then dried again at 50 °C for 24 h. A glass elutriation column with an upward water flow at a velocity of 1.3 cm/s was then used for 15 min. OM was removed with 7.5% sodium hypochlorite (NaClO), leaving the sample in an incubator at 50 °C and 300 rpm for 24 h. The sample was then vacuum filtered with a 20 µm stainless-steel mesh, DS was performed using a 6M ZnCl_2_ solution, and the sample was centrifuged for 5 min at 4900 rpm. The resulting supernatant was filtered again, and the DS process was repeated three times. Braun et al. [11] investigated whether the amount of compost added to agricultural soils was related to a higher concentration of MPs in these soils. For particle extraction, the soil sample was first homogenized with a mortar and then passed through three sieves to classify MPs into mesoplastics, large MPs, and small MPs. A ZnCl_2_ solution was added to each fraction, stirred for 30 min at 200 rpm, and left to settle for 2 h. The supernatant was vacuum filtered, and the filters were dried in an oven at 40 °C.

The efficiency of microplastic (MP) extraction from soils depends significantly on both the incubation time and the type of organic matter removal (ROM) agent used. Stec et al. [61] compared two ROM approaches: 30% H_2_O_2_ for 48 h and 10% KOH for 14 days. In the study, samples were first dried and then subjected to the ROM processes previously mentioned. After treatment, samples were filtered and underwent DS using a saturated NaCl solution, followed by a 24 h settling period. A second filtration step was then carried out and the filters were dried at 60 °C. This entire process was repeated four times. The authors concluded that KOH was more effective for soils with high OM content and that 48 h was sufficient to extract MPs without damaging their structure.

Another study conducted by Seo et al. [23] also emphasized the importance of the removal agent. They tested three different chemical mixtures: Fenton’s reagent (H_2_O_2_/Fe(II)SO_4_ 4:1 at 60 °C), H_2_SO_4_/H_2_O_2_ (1:40 at 60 °C), and piranha solution (H_2_SO_4_/H_2_O_2_ 3:1 at 40 °C). Their results showed that the piranha solution was the most efficient, as it efficiently removed organic matter without degrading the polymers or requiring high temperatures. For DS, the authors avoided high-density solutions such as ZnCl_2_ or NaBr, which can sediment and clog filters after digestion with H_2_SO_4_ and H_2_O_2_. Instead, they used NaCl combined with centrifugation at 2500 rpm for 5 min. The resulting material was filtered through 0.6 µm glass fiber filters and oven-dried at 60 °C for 24 h.

A recent review by Sahai et al. [44] points out that the majority of studies begin by drying soil samples prior to analysis. Specifically, 60.5% of the reviewed works used air drying at 25 °C, while the remainder employed oven drying at temperatures below 75 °C to prevent polymer degradation, 40 °C for 24 h being the most commonly used. The authors also report that a sieving step is frequently included, with 5 mm and 2 mm mesh sizes used in 50% and 18.8% of the studies, respectively. The ROM process typically follows, lasting 24 to 72 h at temperatures between 40 °C and 75 °C. This is followed by a DS step, often assisted by ultrasound to enhance particle separation. Samples are then allowed to settle for 12 to 24 h; in some cases, this sedimentation step is replaced by centrifugation at 2000–3000 rpm. Finally, the supernatant is filtered through membranes with pore sizes ranging from 7 to 20 µm [44].

#### 3.2.2. Sediments

Crutchett and Bornt et al. [29] used the overflow method for MPs separation in sediment samples. First, they performed DS with a ZnCl_2_ solution and then applied overflow, which involves decanting the upper part of the sample (where the separated MPs are located) using the same ZnCl_2_ saline solution. The MPs were collected in another container and the usual filtration process was carried out. Other studies also used ZnCl_2_ for the extraction of MPs from sediments and highlight the importance of using saturated salts with high density such as ZnCl_2_ or NaI, since it is common to find high-density MPs in this type of matrix [84].

Simon-Sánchez et al. [34] mixed beach and marine sediment samples with a saturated NaCl solution and 30% H_2_O_2_, heating the mixture at 50 °C for 20 min with agitation at 200 rpm. Subsequently, the sample was allowed to settle for 1 h at 50 °C and then for another hour at room temperature. The foam formed was transferred to another beaker, mixed again with saturated NaCl and 30% H_2_O_2_, and the heating, agitation, and sedimentation steps were repeated. Finally, the sample was filtered with a glass fiber filter and the filters were dried at 40 °C overnight.

Conventional DS techniques require long periods and are not always efficient; for this reason, a different system has been used to extract MPs from marine sediments. It consists of placing the sample in a stainless-steel cylinder with perforated metal plates, where a flotation liquid and air injection are pumped in, promoting the mixing of the sample with the liquid. The mixture passes through the perforated plates, allowing heavy sediments to accumulate at the bottom while MPs float to the top, where they are suctioned off and passed through sieves of different mesh sizes [77]. Wazne et al. [91] also modified the traditional DS method by replacing the stopcock of the separation funnel with a silicone tube controlled by a Mohr clamp, preventing common clogging issues in DS. The sediment samples were dried for 2 days at 55 °C, followed by DS with ZnCl_2_ in the modified funnel. After agitation, the mixture was allowed to settle for 24 h and was filtered through a 63 µm mesh. Finally, a 30% H_2_O_2_ solution and a 0.05 M Fe(II) solution were added, left to react for 24 h, and then filtered again using the same 63 µm mesh.

Samuels et al. [40] started the extraction by removing moisture from the sample in an oven at 60 °C for 48 h, then passed the sample through a column of sieves with different mesh sizes under agitation for 5 min. For ROM, they added 10% KOH and left the sample in an oven at 60 °C for 48 h. DS is carried out with a saturated NaCl solution, with vigorous agitation at room temperature for 10 min. Finally, the mixture was allowed to settle for 15 min, and the supernatant was filtered. This process was repeated three times. Simon-Sánchez et al. [43] performed an initial treatment of sediment samples with 10% H_2_O_2_ for one week until foaming ceased. A sieving process using a cascade of sieves (from 5 mm to 1 mm) was then carried out and the sample was left to settle for one week. DS was then performed with ZnCl_2_, and 5% SDS was added as a surfactant. This is followed by enzymatic digestion using protease, cellulase, and Viscozyme, an oxidative catalysis with Fenton’s reagent, and a second DS with ZnCl_2_. Finally, a new sieving process (from 1000 µm to 10 µm) was performed to classify the particles into large and small fractions, which were then dried in an oven at 50 °C. Bailey et al. [88] analyzed sediment samples from bays, performing an initial wet sieving using a 45 µm filter, a ROM with H_2_O_2_, and a DS with lithium polytungstate (Li_2_WO_4_) due to its high density and low environmental impact. Finally, they performed filtration using a 1.2 µm glass filter. Other reviewed studies on MPs extraction in river sediment samples, highlighting various DS and ROM procedures, include: Na_2_WO_4_·2H_2_O and 30% H_2_O_2_ [92], ZnCl_2_ and 30% H_2_O_2_ [81], and ZnCl_2_ and 30% H_2_O_2_ + iron solution [80]. All of them end with filtration using different types of filters: the first study used a 5 µm silver filter, the second study a 0.7 µm glass filter, and the last study two sieves of 1 mm and 0.3 mm in diameter.

#### 3.2.3. Wastewater

In wastewater samples, Shalumon et al. [33] analyzed washing machine effluents. They added acetone to the sample and stirred it. The mixture was then poured into a beaker with distilled water and finally filtered with a 0.2 µm glass filter. Other authors, such as Garzón-Vidueira et al. [71], used an oil-based extraction method with sunflower oil. In this case, the sample was placed in a separation funnel along with a NaCl solution, stirred, and left on ice for 30 min. Then, sunflower oil was added, and the mixture was stirred for 3 min. After this time, the aqueous phase was discarded, and the oil phase was vacuum filtered, after which the filters were placed in the oven. It is worth noting that in wastewater samples with moderate OM content, a prior digestion with 30% H_2_O_2_ for 2 h at 70 °C was carried out. However, in samples with low OM content, this step was omitted to avoid affecting MPs.

In brief, in samples with moderate OM content (Section 3.2), studies on agricultural soils generally combine the ROM and DS stages. The process begins with drying the samples (usually at 40 °C for 24–48 h) followed by sieving (between 5 mm and 40 µm). For ROM, the most commonly used method is 30% H_2_O_2_, although KOH, NaClO, or Fenton’s reagent are also used, depending on the %OM present. In the DS stage, solutions such as NaCl, ZnCl_2_, or SPT are commonly used, often combined with centrifugation or sedimentation. Alternatives such as the use of vegetable oils or elutriation have also been explored. Finally, filtration is performed using filters ranging from 0.6 to 20 µm, which are then dried in an oven. The procedure for sediment samples is very similar. DS techniques are used with ZnCl_2_, NaCl, or NaI solutions, with ZnCl_2_ standing out for its effectiveness in recovering high-density MPs. ROM is typically carried out using 30% H_2_O_2_, Fenton’s reagent, or KOH. Pretreatment usually includes drying at 40–60 °C for 24–48 h and sieving, followed by several cycles of sedimentation, agitation, or centrifugation. Final filtration is performed using filters ranging from 0.7 to 63 µm. Alternative approaches are also employed, such as the overflow method, systems with air injection, and columns with modified valves to prevent clogging. In wastewater samples, ROM is typically carried out using 30% H_2_O_2_, while for DS, solvents such as acetone and techniques based on vegetable oils are used in combination with saline solutions and subsequent filtration.

### 3.3. Samples with Low OM Content (Atmospheric Samples, Beach Sand, Seawater, and Fresh Water)

#### 3.3.1. Atmospheric Samples

For atmospheric samples, Edo et al. [30] analyzed environmental samples from a protected area over multiple seasons, both wet and dry, for comparison. To achieve this, they placed a metal filter in a particle collector. After removal, the filters were treated with 33% H_2_O_2_ at 60 °C for 24 h to remove OM and the particles were identified with a stereomicroscope without the need to carry out the flotation process. Pescoso-Torres et al. [93] also used a particle collector to analyze MPs deposition at two different sites (rural and urban) in Cienfuegos, Cuba. After the sampling period, filters were treated with 30% H_2_O_2_ at 45 °C for 8 days. A DS with NaCl was then performed, followed by filtration and drying of the filters at 45 °C for 24 h.

#### 3.3.2. Beach Sand

For beach sand samples, Simon-Sánchez et al. [34] added a saturated NaCl solution to the sample and agitated it for 20 min at 200 rpm. The supernatant was subsequently filtered with a 0.7 µm glass fiber filter, and the filters were dried at 40 °C overnight. Another study by Ghanadi et al. [42] focused on MPs extraction from sandy sediments in coastal environments. Samples were first dried at 60 °C and sieved through a 500 µm mesh, retaining the <500 µm fraction. DS was then performed using a saturated CaCl_2_ solution, followed by 15 min of sonication and 15 min of centrifugation at 1800 rpm. The supernatant was filtered using a 5 µm polycarbonate (PC) filter. However, although Zala et al. [94] perform a ROM using 30% H_2_O_2_ along with a DS using NaCl, beach sand samples are generally considered a clean matrix, and most studies omit this step due to their low organic matter content [26,95,96,97,98].

#### 3.3.3. Seawater

Ghanadi et al. [42] also extracted MPs from seawater samples from coastal environments. First, the samples were filtered using a 15 µm PC filter, onto which 30% H_2_O_2_ was added to remove OM. The mixture was then re-filtered with the same 15 µm PC filter and subjected to DS with CaCl_2_. Finally, a final filtration was performed with a 5 µm filter. A recent study by Murphy-Hagan et al. [37] analyzed MPs extraction in urban water bodies from coastal basins. The samples were first filtered with a 250 µm metal sieve, removing particles larger than 5 mm. The remaining samples underwent digestion with 30% H_2_O_2_ and Fenton’s reagent, using an ice bath if the temperature exceeded 60 °C. Finally, the samples were filtered again with a 250 µm sieve. Another recent study conducted by Zhou et al. [74] analyzed the extraction of MPs from the Indian Ocean. First, a wet sieving was performed using a 5 mm stainless-steel sieve to remove larger particles. Next, a second sieving step was carried out using 1000 µm, 350 µm, 250 µm, and 150 µm sieves. Subsequently, the ROM process was applied using Fenton’s reagent, followed by DS with NaCl and ending with a filtration using a 0.45 µm nitrocellulose membrane.

#### 3.3.4. Fresh Water

Queiroz et al. [73] applied an MPs extraction protocol to sediment samples from fresh water. The process consisted of five stages: first, the samples were dried at 60 °C for 4 h, then a DS was performed using a NaCl solution assisted by centrifugation at 3500 rpm for 10 min. The supernatant was filtered using a 0.7 µm glass filter, and subsequently, the filters were dried at 60 °C. Finally, ROM was performed by adding a few drops of 30% H_2_O_2_ at 60 °C for 24 h, after which the filters were analyzed. For their part, Jaikumar et al. [99] compared the presence of MPs in seawater and freshwater environments, and for the extraction of freshwater samples, they employed an enzymatic–oxidative process for the ROM, followed by DS with ZnCl_2_.

To summarize Section 3.3, in atmospheric samples, MPs are collected on filters using particle collectors. A digestion with H_2_O_2_ (30–33%) is then applied at temperatures between 45 and 60 °C for periods ranging from 24 h to 8 days. In some cases, a DS with NaCl is also performed, followed by filtration and drying of the filters. In beach sand samples, due to their low OM content, many studies omit the ROM step. The typical protocol involves DS using NaCl or CaCl_2_ solutions, sometimes combined with sonication or centrifugation to improve efficiency. In some cases, 30% H_2_O_2_ is used if a significant amount of OM is present, although this is not common. The separated fractions are filtered using filters ranging from 0.7 to 5 µm and then dried before analysis. In fresh and seawater samples, the most commonly used methods include an initial sieving step, followed by DS with NaCl or ZnCl_2_, often in combination with centrifugation. ROM can be performed with 30% H_2_O_2_ or through more complex protocols combining enzymatic and oxidative digestion, especially in samples with some OM content. Finally, MPs are recovered through filtration using glass fiber filters or membranes with pore sizes down to 0.7 µm and then dried.

### 3.4. Samples with Very Low OM Content (Drinking Water)

The digestion step to dissolve particulate matter is used only in some studies to eliminate impurities and optimize filtration. However, these samples have minimal OM, so most studies skip this step [79].

#### 3.4.1. Tap Water

Pivokonsky et al. [100] applied WPO at 75 °C for digestion, followed by double filtration: first with a 5 µm filter and then with a 0.2 µm polytetrafluoroethylene (PTFE) filter. Filters were then dried in an oven at 30 °C for 30 min. On the other hand, Shruti et al. [101] and Zhang et al. [102] did not consider a digestion step necessary. In the first case, samples were filtered directly using a 0.22 µm nitrocellulose filter and left to dry at room temperature. In the second one, filtration was performed with a 0.45 µm filter.

#### 3.4.2. Bottled Water

For bottled water samples, Oßmann et al. [103] performed digestion using an EDTA solution, followed by DS with an SDS solution and filtration with a 0.4 µm PC filter. Conversely, most studies skipped the digestion step and carried out direct filtration using different types of filters: a 1.5 µm glass filter [104], a 3 µm gold-coated PC filter [105], and 0.45 µm nitrocellulose filters [106,107]. A more recent study by Hossain et al. [108] also omitted the digestion step, filtering the water samples directly using a 5 µm nitrocellulose filter.

To sum up Section 3.4, in drinking water (tap and bottled water) samples, due to their very low OM content, the digestion process is generally unnecessary. Instead, MPs extraction is primarily performed through direct filtration using filters ranging from 0.2 to 5 µm. In cases where digestion is carried out, WPO methods or EDTA solutions are used, and occasionally a DS step is included, followed by filtration and drying of the filters in an oven.

It is essential to highlight that extraction processes are lengthy, tedious, meticulous, and crucial processes for obtaining reliable results. To minimize plastic contamination, it is recommended to avoid plastic containers, such as tupperware, tubes, jars, plates, or bags throughout the entire process. Instead, glass materials should be used, such as watch glasses or glass Petri dishes, and samples should be covered with aluminum foil rather than plastic. Additionally, it is critical to keep samples covered to prevent contamination from airborne particles and to filter all reagents used, including saturated saline solutions for DS and deionized water, to prevent cross contamination. To ensure good recovery of MPs, it is advisable to repeat certain procedures and thoroughly rinse the walls of the materials used with suitable solvents, avoiding the loss of MPs that may have adhered [30,31,33,58,100].

## 4. Conclusions

This review responds to the need to establish standardized and effective methods for the extraction of microplastics (MPs) from various environmental matrices due to the growing concerns about their presence and the challenges involved in the extraction process depending on the type of sample.

Common steps are observed across protocols, including initial drying, sieving, removal of organic matter (ROM), density separation (DS), filtration, and drying of the filters. Regarding reagents, the most commonly used for ROM is H_2_O_2_ at 30% due to its effectiveness and low impact on polymers. However, in samples with high organic content, it is often combined with Fenton’s reagent or enzymatic digestion. For DS, NaCl is the most frequently used because of its low cost and low toxicity, especially for low-density plastics. ZnCl_2_ is preferred for recovering high-density polymers, as it can be reused multiple times without losing efficiency, despite its higher cost.

Depending on the matrix type, some more specific aspects must be considered: sludge and compost samples, with high organic content (50–70%), require more intensive chemical digestion and DS using NaCl, ZnCl_2_, or SPT. Soils and sediments, with moderate organic content (0.5–5%), typically require H_2_O_2_ or Fenton’s reagent along with NaCl or ZnCl_2_, often assisted by centrifugation. In less complex matrices, such as beach sand, freshwater, seawater, and atmospheric samples, ROM is usually omitted, and simple DS or direct filtration is preferred. For drinking water, where the organic content is extremely low (<0.001%), extraction is limited to direct filtration. Although significant progress has been made, this review highlights the urgent need to standardize protocols that balance efficiency, cost, time, and chemical safety. Looking ahead, developing methods that are more sustainable, less aggressive, and better adapted to different matrices will be essential to ensure effective recovery and the study of MPs in order to preserve the environment.

In conclusion, although there is currently no universal and optimal extraction method, the reviewed protocols have shown high recovery rates, provided they are properly adapted to the type of sample being analyzed. Methods that use ZnCl_2_ for DS and H_2_O_2_ (either alone or in combination with Fenton’s reagent) for ROM exhibit particularly effective performance. However, the variability in recovery efficiency, the need to optimize reagents and operational parameters such as temperature and extraction time, as well as the compatibility with subsequent identification techniques, remain important limitations to consider.

## Figures and Tables

**Figure 1 molecules-30-03178-f001:**
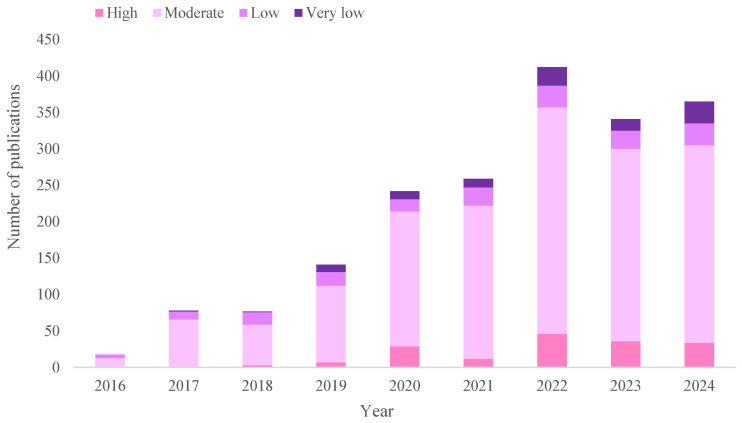
Number of publications on MPs extraction from environmental matrices according to their %OM from 2016 to 2024, basing on the database of Web of Science.

**Figure 2 molecules-30-03178-f002:**
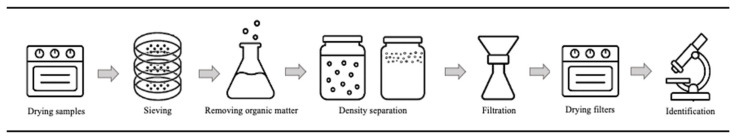
Summary of the commonly used steps in the extraction of MPs from environmental samples.

**Figure 3 molecules-30-03178-f003:**
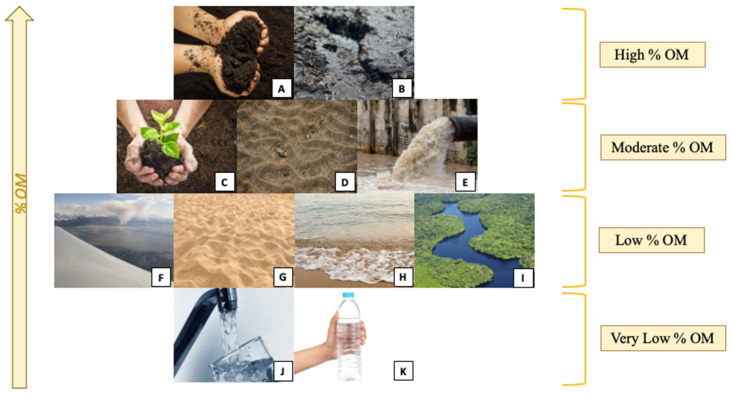
Samples from different environmental matrices depending on their %OM, where (**A**) compost, (**B**) sludge, (**C**) agricultural soil, (**D**) sediments, (**E**) wastewater, (**F**) atmospheric sample, (**G**) beach sand, (**H**) seawater, (**I**) fresh water, (**J**) tap water, (**K**) bottled water.

## Data Availability

Data will be available on request.

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
