# Peer review of "Extraction Methods of Microplastics in Environmental Matrices: A Comparative Review"

_molecules, 2025, doi:10.3390/molecules30153178_

Round 1
Reviewer 1 Report
Comments and Suggestions for Authors
The submitted review article provides an excellent overview of the methods used to isolate microplastic particles from environmental sources with varying levels of organic matter content. The summary in Table 1 is useful.
- It is clear that the focus is solely on extraction and not on subsequent characterization. However, it would be helpful to mention any limitations of the extraction methods imposed by the subsequent characterization method.
- When evaluating the protocols, it is important to mention whether they were established with spiked samples, whether they were stepwise optimized, and whether samples containing mixtures or single polymers were studied. This information could be summarized in a table.
- Could Figure 3 be expanded to include a scheme of the optimized treatment?
- What is the authors' conclusion regarding the efficacy of the methods?
Author Response
Reviewer 1:
[The submitted review article provides an excellent overview of the methods used to isolate microplastic particles from environmental sources with varying levels of organic matter content. The summary in Table 1 is useful.]
Thank you very much for considering our manuscript and for your valuable comments. The article has been significantly improved thanks to the incorporation of the additional information requested by the reviewers.
[It is clear that the focus is solely on extraction and not on subsequent characterization. However, it would be helpful to mention any limitations of the extraction methods imposed by the subsequent characterization method.]
Just before the new section 2.2., entitled The correlation between OM amount and abundance of MPs, we have added some new paragraphs to mention more limitations and considerations of the extraction methods imposed by the subsequent characterization methods.
In addition, the revised version of the paper includes additional techniques beyond LDIR, such as Raman spectroscopy and morphological analysis.
[When evaluating the protocols, it is important to mention whether they were established with spiked samples, whether they were stepwise optimized, and whether samples containing mixtures or single polymers were studied. This information could be summarized in a table.]
According to the reviewer, we have added information about the type of the sample employed when evaluating the protocols. In the revised version, Table 2 includes this information.
For each reference, information on the sample type has been included. We have added asterisks in two different colours: a green asterisk (*) means that the polymer sample in that item was spiked while an orange asterisk (*) signifies that the polymer was extracted from real environmental samples. When both asterisks appear, it means that the polymer was first used in spiked samples to optimize the method and then extracted from real environmental samples.
[Could Figure 3 be expanded to include a scheme of the optimized treatment?]
We sincerely thank the reviewer for this thoughtful and constructive suggestion. Regarding the proposal to expand Figure 3 to include a schematic representation of the optimized treatment, we gave this idea careful consideration. However, we found it extremely challenging to develop a unified and representative scheme due to the high methodological variability among the studies reviewed, even within the same organic matter content group (High %OM, Moderate %OM, Low %OM, Very Low %OM).
In particular, the reviewed protocols differ considerably in terms of digestion reagents, density separation media, reaction times, and temperatures, even for samples with comparable OM content. Therefore, attempting to consolidate these diverse approaches into a single schematic could inadvertently oversimplify or misrepresent the complex and heterogeneous nature of the methodologies reported.
We truly appreciate the reviewer’s understanding and, once again, thank them for their insightful feedback.
[What is the authors’ conclusion regarding the efficacy of the methods?]
Thank you for your comment. In the final paragraph of the Conclusions section, we have added specific remarks on the effectiveness and limitations of the methods discussed.
Reviewer 2 Report
Comments and Suggestions for Authors
All the comments are in the attached file.

Author Response
Reviewer 2:
Thank you very much for considering our manuscript and for your valuable comments. The article has been significantly improved through the incorporation of the additional information requested by the reviewer. Moreover, the manuscript has been reviewed and edited by a native English speaker to improve the language and clarity.
[You need to provide the keywords used for the search, as well as their combinations, together with the databases interrogated. Also, you need to justify the time interval of the searcher]
As suggested by the reviewer, we have included the keywords and the search combinations used. The literature search was conducted in the Web of Science database, covering the period from 2016 to 2024.
[Here could start a sub-chapter, 2.1 removal of OM (previous line 114)]
In the revised version of the manuscript, a new sub-section has been added: 2.1. Removal of organic matter (ROM).
[You need to rephrase this, not clear, while the grammar has some problems (previous line 118)] [You need to rephrase this, the sample and not the reagent is the main subject here (line 122)].
Both sentences have been rephrased and reviewed by a native speaker.
["their" works better (Table 2)].
This correction has been implemented as requested.
[Here could start a second sub-chapter 2.2 The correlation between OM amount and abundance of MPs (previous line 219)].
We agree with the reviewer’s comment. Thank you once again for your valuable feedback. In the revised version of the manuscript, we have included a new sub-section: 2.2. The correlation between OM amount and abundance of MPs.
[From line 221 till 286 there is nothing related to the title of the chapter. All this information should be relocated to chapter 2, prior to the ways of removing OM. More, this information should be split - the amount of OM at the beginning of the chapter, its removal, in the rest of the chapter.]
After introducing the new sub-sections (2.1 and 2.2) and modifying the beginning of Section 3 (previously line 287), the content from lines 221 to 286 has been reorganized accordingly. We believe that these modifications adequately address the reviewer’s concerns. Thank you again for your valuable comments.
[Here is the start of chapter 3 (line 287)].
We agree with the reviewer. This now marks the starting point of Section 3 in the revised manuscript.
[No citation, although "most studies" was mentioned (previous line 428)].
You are right. We had omitted the reference. We have now added the appropriate citation [44] (Sahai et al.) in the revised manuscript.
Round 2
Reviewer 1 Report
Comments and Suggestions for Authors
Thank you for implementing my comments.
Reviewer 2 Report
Comments and Suggestions for Authors
Complying with the reviewers’ recommendations/suggestions, the authors rendered this paper publishable as is.